

# Prediction of Ross River virus incidence in Queensland, Australia: building and comparing models

Wei Qian[1], David Harley[1], Kathryn Glass[2], Elvina Viennet[3,4] and Cameron Hurst[5,6]

[1] The University of Queensland, UQ Centre for Clinical Research, Herston, Queensland, Australia
[2] Research School of Population Health, Australian National University, Acton, Australian Capital Territory, Australia
[3] Clinical Services and Research, Australian Red Cross Lifeblood, Kelvin Grove, Queensland, Australia
[4] Institute for Health and Biomedical Innovation, School of Biomedical Sciences, Queensland University of Technology, Kelvin Grove, Queensland, Australia
[5] Molly Wardaguga Research Centre, Charles Darwin University, Brisbane, Queensland, Australia
[6] Department of Statistics, QIMR Berghofer Medical Research Institute, Brisbane, Queensland, Australia

Corresponding author
Cameron Hurst,
cameron.hurst@cdu.edu.au,
Cameron.Hurst@qimrberghofer.edu.au

## ABSTRACT

Transmission of Ross River virus (RRV) is influenced by climatic, environmental, and socio-economic factors. Accurate and robust predictions based on these factors are necessary for disease prevention and control. However, the complicated transmission cycle and the characteristics of RRV notification data present challenges. Studies to compare model performance are lacking. In this study, we used RRV notification data and exposure data from 2001 to 2020 in Queensland, Australia, and compared ten models (including generalised linear models, zero-inflated models, and generalised additive models) to predict RRV incidence in different regions of Queensland. We aimed to compare model performance and to evaluate the effect of statistical over-dispersion and zero-inflation of RRV surveillance data, and non-linearity of predictors on model fit. A variable selection strategy for screening important predictors was developed and was found to be efficient and able to generate consistent and reasonable numbers of predictors across regions and in all training sets. Negative binomial models generally exhibited better model fit than Poisson models, suggesting that over-dispersion in the data is the primary factor driving model fit compared to non-linearity of predictors and excess zeros. All models predicted the peak periods well but were unable to fit and predict the magnitude of peaks, especially when there were high numbers of cases. Adding new variables including historical RRV cases and mosquito abundance may improve model performance. The standard negative binomial generalised linear model is stable, simple, and effective in prediction, and is thus considered the best choice among all models.

## INTRODUCTION

Ross River virus (RRV) is a mosquito-transmitted *Alphavirus* that causes arthritis, rash, and constitutional symptoms of fever, fatigue, and myalgia (*Liu, Tharmarajah & Taylor, 2017*; *Harley, Sleigh & Ritchie, 2001*). It is the most common human arboviral infection in Australia. Between 1993 and 2020, almost half of the nationally notified cases (49.0%, 63,880/130,271) occurred in Queensland with a total of 43,699 cases reported from 2001 to 2020 (*Qian et al., 2021*; *Australian Department of Health, 2021*). Outbreaks of RRV have occurred regularly in Queensland over the past two decades (*Qian et al., 2021*). Timely prediction of outbreaks based on epidemiological models is necessary for disease prevention and control. Transmission of RRV is influenced by mosquito abundance, reservoir host populations, climate and weather, geographical exposures, and socio-economic indices (*Qian et al., 2020*; *Murphy et al., 2020*; *Tong et al., 2008*). Weather patterns may influence RRV transmission differently at different times and in different parts of Queensland (*Qian et al., 2020*). The dominant vector and reservoir host species in coastal and inland areas differ (*e.g.*, *Culex annulirostris* breeds in freshwater habitats, *Aedes vigilax* breeds in mangroves and salt marshes, and *Aedes notoscriptus* breeds in artificial containers) (*Stephenson et al., 2018*; *Harley et al., 2000*). In Queensland, temperature, rainfall, relative humidity, and high tide are possible predictors of RRV transmission in coastal regions, while temperature, or relative humidity and rainfall are associated with inland RRV activity (*Tong & Hu, 2002*; *Bi & Parton, 2003*). The number of exposures and the complex relationships between these exposures (*Ng et al., 2014*; *Gatton, Kay & Ryan, 2005*) make RRV modelling challenging.

Many studies have applied epidemiological models to predict RRV cases, incidence, or outbreaks, with exposures including mosquito abundance, rainfall, temperature, tidal height, humidity, and river flow (*Qian et al., 2020*). Half of these studies employed generalised linear models, which are straightforward and simple, and hence commonly used in predicting infectious diseases. Other approaches such as time-series models, spatial and temporal models, hurdle models, and Generalised Additive Models (GAMs) were also used. A few studies have published data related to model performance. Performance accuracies varied from 63.0% to 100.0%, sensitivities and specificities varied from 0.0 to 1.0, overall agreements varied from 75.8% to 88.5%, and true positives varied from 0.0% to 100.0% (*Hu et al., 2006*; *Woodruff et al., 2002*; *Woodruff et al., 2006*; *Jacups, Whelan & Harley, 2011*; *Pelecanos, Ryan & Gatton, 2010*; *Gatton, Kay & Ryan, 2005*; *Koolhof, Bettiol & Carver, 2017*; *Ng et al., 2014*). Most weather and climatic exposures used in previous studies (*e.g.*, rainfall, temperature, vapour pressure, and evaporation) have been found to have positive effects on disease transmission, which means that increases of these exposure values are related to an increase in disease incidence. High relative humidity generally reduces RRV incidence. Other factors include mosquito abundance, tidal height, and river flow (*Qian et al., 2020*). Stepwise variable selection was often used in developing RRV predictive models (*Koolhof et al., 2021*). Regional analysis was suggested but rarely used in previous studies (*Ng et al., 2014*). Studies assessing predictive model comparison are lacking.

Selecting important predictors can improve our understanding of disease transmission. A model building process with detailed considerations of lagged predictors, variable selection, and validation, and a head-to-head comparison of the models will allow assessment of suitable model frameworks for future prediction of RRV. We applied 10 models and compared their model fit and performance in predicting RRV incidence. The Poisson and negative binomial generalised linear models were implemented to fit the distribution of RRV cases. Zero-inflated Poisson and Zero-inflated negative binomial models were developed due to the excess zeros caused by relatively small spatial and temporal resolution. Generalised additive models with and without zero-inflated Poisson or zero-inflated negative binomial models were also used. A variable selection strategy to screen the most important variables for prediction was developed and regional analysis was used.

By building and comparing these models, we aimed to (1) identify the most robust approach for predicting RRV incidence with good model fit among generalised linear models, zero-inflated models, and GAMs; and (2) develop a model building process that includes an effective variable selection strategy to build models with both excellent predictive performance and clear epidemiological logic in different regions. The results and the strategy provide important guidance for mosquito-borne disease modelling.

## MATERIALS & METHODS

### Data and study design

This is an ecologically designed study involving modelling and predicting RRV incidence using aggregated notifications, and climatic, geographical, and socio-economic exposures. A confirmed RRV case is defined as virus isolation, virus detection by nucleic acid testing, or IgG seroconversion or a significant increase in IgG antibody level (*Australian Department of Health, 2016*). A probable RRV case requires detection of RRV IgM and IgG, except when IgG has also been detected more than three months previously. Both confirmed cases and probable cases are notified (*Australian Department of Health, 2016*).

Notified RRV cases at Statistical Area Level 3 (SA3) in Queensland between January 1, 2001, and December 31, 2020, were acquired from the Queensland Department of Health. These SA3 areas generally have a population between 30,000 and 130,000 (*Australian Bureau of Statistics, 2016*). The notification data were in the form of daily de-identified records by date of report. The total numbers of historical cases at SA3 level from 1991 to 2000 in Queensland were also acquired from the Queensland Department of Health, to estimate the population immunity level. An annual human population estimate in each SA3 area during the study period was obtained from the Australian Bureau of Statistics (*Australian Government, 2019c*). The susceptible population was calculated as the total population minus the historical RRV infected population. Incidence rates were calculated by dividing the number of cases by the total population of a given area.

Daily weather data for the period were acquired from the Australian Bureau of Meteorology for rainfall, temperature, relative humidity, evaporation, evapotranspiration, and vapour pressure, including the maximum, minimum and average values. Extreme

events of flooding, bushfire, and long-term climatic indices such as La Niña episodes, El Niño events, and Southern Oscillation Index (SOI) occurring during the study period were sourced from the Australian Bureau of Meteorology (*Australian Government, 2019b*).

The Socio-economic Index for Areas (SEIFA) was collected from the Australian Bureau of Statistics (*Australian Government, 2019c*). The Accessibility/Remoteness Index of Australia, known as the ARIA score, was acquired from the Hugo Centre for Population and Housing at the University of Adelaide (*Hugo Centre for Population and Housing, 2020*).

Monthly Normalized Difference Vegetation Index (NDVI) was collected from the Bureau of Meteorology website. Other geographical exposures including lake areas, reservoirs, wetlands, land use data and elevation were collected from the Queensland Spatial Catalogue and Geoscience Australia website (*Queensland Government, 2021*; *Australian Government, 2019a*). Digital elevation data were obtained from the Geoscience Australia website and were incorporated with other data at SA3 level through Geographical Information Systems.

Exposures were extracted and summarised as weekly data in SA3 areas. In this study, the years start from January and the weeks start from Monday. When aggregating daily data to weekly data, if not all the daily data in a week were available, the average values of non-missing data were calculated for the week. Linear extrapolation was used to estimate the human population before June 2001. When estimating weekly data by annual or monthly data (*e.g.*, human population and NDVI), data were converted to daily data by linear interpolation, then were summarised as weekly data by calculating an average for continuous or a median for ordinal data.

The spatial climatic and geographical data collected in map format were transformed and aggregated in ArcGIS first. If data were missing in some SA3 areas, Nearest Neighbour Interpolation was applied if adjacent grid points existed, otherwise Inverse Distance Weighting was used for spatial interpolation (*Li & Heap, 2014*).

For ARIA and SEIFA indices, the survey data in 2011 were used to extrapolate data before 2014, and survey data in 2016 were used to extrapolate data after 2015. Data only available in Statistical Area Level 2 (SA2) areas were aggregated into SA3 areas by calculating population-weighted averages (for SEIFA) or medians (for ARIA). If the median of the ARIA score was not an integer, the ARIA score in the largest SA2 area was chosen.

To better predict RRV incidence based on varying climatic patterns of different areas, the Statistical Area Level 3 (SA3) areas were grouped into three regions according to Thermal Climate Zone Classification (*Australian Government, 2022*; *Stewart & Oke, 2009*) to build additional models in these regions. More detail on the data sources and data collation processes is provided in Text S1.

## Variable selection

Before building the model, we developed a variable selection process based on the purposeful selection of covariates strategy proposed by *Hosmer, Lemeshow & May (1999)* to obtain a parsimonious variable set which fitted the hypotheses of the models and had strong epidemiological and statistical correlations with RRV activity. In this study, exposures were the potential influencing factors of RRV transmission regardless of lags. The impact of exposures on RRV transmission may lag by several weeks. We referred to

the exposures at specific lags, including lag zero which represents the current value of an exposure, as predictors (or lagged variables). For example, temperature was an exposure, and maximum weekly temperature at lag 5, which was at 5 weeks earlier, was a specific model predictor.

Optimal weekly lags of temperature, range of temperature, relative humidity, rainfall, vapour pressure, evaporation, evapotranspiration, SOI, El Niño events, La Niña events, NDVI, bushfires and flood events were selected based on the maximum positive and minimum negative values of the cross-correlation function, which showed the correlation between RRV cases and exposures at different lags (*Curriero, Shone & Glass, 2005*). Due to the possible time periods of mosquito breeding, transmission to reservoir hosts, the incubation period in humans and lags for Ross River virus reporting, we considered that lags greater than one year would not plausibly have meaning for associations of exposures and RRV incidence.

Then, a univariable analysis was implemented to exclude predictors with *p* value greater than 0.1 or standardised regression coefficients less than 0.1. These predictors were considered as having a weak impact on RRV incidence. Spearman correlation evaluates the monotonic relationship between two variables. Predictors with absolute value of Spearman correlations greater than 0.9 were highly correlated with each other (*Akoglu, 2018*). One of the two correlated predictors was considered as redundant and was excluded. The Variance Inflation Factor (VIF) was used to calculate the inflation of coefficient variance and address the magnitude of multicollinearity in models. Predictors with high VIF values were removed to reduce redundancy and to avoid possible problem caused by multicollinearity (*e.g.*, distorted statistical significance or predictor coefficients which could misinterpret the effect of predictors on RRV transmission). A repeated backward stepwise screen and reassessment process was used to screen predictors improving model fit in predicting RRV incidence with tolerable multicollinearity. Predictors with VIF greater than five were removed (*Menard, 2002*). The predictor with lowest Bayesian Information Criterion (BIC) increment was removed in each iteration until the total BIC increment reached 0.5%. The predictors were then considered in the model one at a time, and those predictors with VIF values less than ten and BIC decrements greater than 0.1% were included. We continued this process until no predictor was excluded or included, two consecutive iterations returned the same variable set, or it reached 11 iterations. Initially, this process was applied to the climatic variable set, and to the geographical and socio-economic variable sets separately, then, we applied to all variables selected from the two sets. Finally, predictors with VIF greater than five were excluded. The remaining predictors were selected for model building. Further details of this process are shown in Fig. 1.

## Model building

We built models for longitudinal data involving exposures at different lags, which were selected in different data sets. As such, cross-validation methods based on random re-sampling (such as K-fold) were not appropriate as they may result in a look-ahead bias where later data can be used to predict earlier data (*Bergmeir & Benítez, 2012*). Instead, a method that furnishes contiguous blocks of data was required due to the lagged nature of
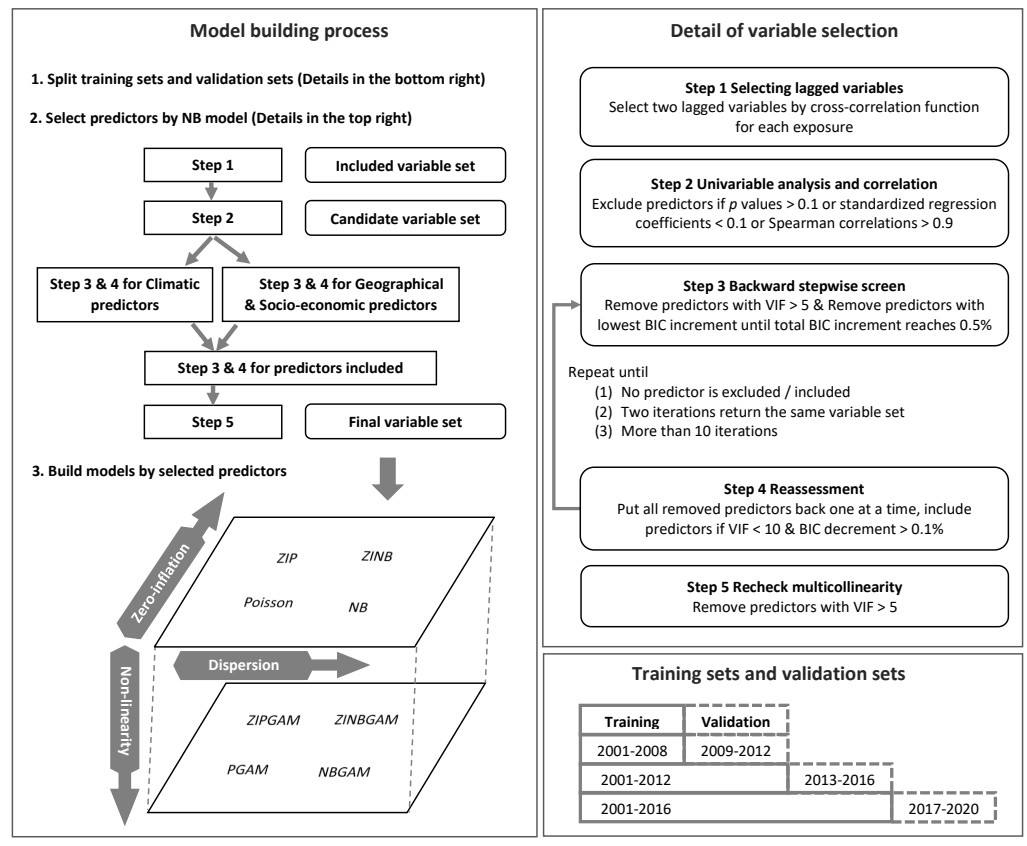

**Figure 1  Process of model building.**

predictors in the model. Similarly, using data from some SA3 areas as training sets and the others as validation sets could lead to bias because the RRV transmission cycle and disease incidence varies in different areas. Instead, a time-series cross-validation approach was applied (*Ramos & Oliveira, 2016*). Data between 2001–2008, 2001–2012, and 2001–2016 were used as the three training sets. The corresponding validation sets included data from 2009–2012, 2013–2016, and 2017–2020, respectively.

Ross River virus notifications were rare counts as the weekly SA3 area data used were at relatively high spatial and temporal resolution. Poisson and negative binomial distributions were appropriate for modelling these data. There are excess zeros according to the distribution plots of RRV cases (Fig. S1). Some predictors may have complex associations with each other and with RRV incidence; non-linear models were used to address the possible non-linear effect of exposures on RRV activity. So, we conducted negative binomial-based models which highlighted the effect of statistical over-dispersion compared with Poisson-based models; zero-inflated models were applied to analyse the effect of zero-inflation caused by excess zeros, and non-linear models were used to account for non-linearity. Hurdle models were not used because zero data can only be generated from the excess zero part of a hurdle model and positive count data is produced from the non-zero part. One of the assumptions of zero-inflated models is that zero data can be

structural zeros from the zero part or sampling zeros from the non-zero part of the model, which is a better conceptual fit to RRV data than that of the hurdle models.

As the standard negative binomial generalised linear model was simple, stable, and fitted the data well in preliminary analyses, it was used to select the final variable set for all models. In preliminary analyses, the same model was used to select predictors and to build models to access the possible impact of the method used for variable selection on final model performance. These variable sets were then used in all models for comparison. The ten models were: standard Poisson generalised linear model (Poisson), standard Negative Binomial generalised linear model (NB), Zero-Inflated Poisson model with constant in zero part (ZIP), Zero-Inflated Poisson model with exposure as a regressor in zero part (ZIPe), Zero-Inflated Negative Binomial model with constant in zero part (ZINB), Zero-Inflated Negative Binomial model with exposure as a regressor in zero part (ZINBe), Poisson Generalised Additive Model (PGAM), Negative Binomial Generalised Additive Model (NBGAM), Zero-Inflated Poisson Generalised Additive Model (ZIPGAM), and Zero-Inflated Negative Binomial Generalised Additive Model (ZINBGAM). Details of the models are provided in Text S1. To identify important predictors and test the model building process in different regions, the model building process was conducted in each region separately, and then in the full dataset (all Queensland) to address the overall model performance.

## Model performance

Model fit was assessed by Akaike information criterion (AIC) (*Akaike, 1974*), BIC, and Hannan-Quinn Information Criterion (HQIC). The AIC does not take sample size into consideration whereas BIC and HQIC use sample size as a penalty weight (*Claeskens & Hjort, 2008*). Among BIC and HQIC, BIC is stricter in penalising sample size. Because a large number of predictors were screened in this study, a parsimonious model having fewer predictors was preferable. So, BIC was used to optimise the model, and to evaluate the model fit if the information criteria were inconsistent.

The AIC is given by:

$$\text{AIC} = -2\log(\text{L}) + 2k$$

where $L$ is the maximum likelihood of the data and $k$ represents the number of the unknown parameters in the model.

The BIC is given by:

$$\text{BIC} = -2\log(\text{L}) + \log(n) * k$$

where $n$ is the sample size.

The HQIC is given by:

$$\text{HQIC} = -2\log(\text{L}) + \log(\log(n)) * k.$$

As the information criterion statistics could only be generated for the training sets, measures to gauge model predictive performance in the validation sets were required. Three measures of relative residuals were applied to assess the model accuracies in both

training and validation sets: Mean Square Error (MSE), Root Mean Square Error (RMSE), and Mean Absolute Error (MAE). The more robust MAE was used if the relative residuals were inconsistent. The functional forms of these relative residuals are:

$$\text{MSE} = \frac{\sum_{i=1}^{n}(X_i - \hat{X_i})^2}{n}$$

$$\text{RMSE} = \sqrt{\frac{\sum_{i=1}^{n}(X_i - \hat{X_i})^2}{n}}$$

$$\text{MAE} = \frac{\sum_{i=1}^{n}|X_i - \hat{X_i}|}{n}$$

where $X$ is the predicted or real value of RRV cases and $n$ is the sample size.

All analyses and figures were performed and produced with "MASS" (*Ripley et al., 2021*), "pscl" (*Jackman et al., 2020*), "mgcv" (*Wood, 2021*), and "gamlss" (*Stasinopoulos et al., 2021*) in R 4.1.0 (R Core Team, Vienna, Austria, 2021) (*R Core Team, 2021*) and ArcMap 10.7.1 (Environmental Systems Research Institute, Redlands, CA, USA, 2019) using GCS_GDA_1994 Geographic Coordinate Systems. R code of this study is provided in Text S2. The study was approved and informed consent was waived by the University of Queensland Human Research Ethics Committee A (2019/HE002772).

# RESULTS

Regions of Queensland have different ecological and climatic patterns, vector species, and host populations, which influence RRV transmission. As shown in Fig. 2, 14 SA3 areas have a hot humid summer (abbreviated as "Hot" region); 61 have a warm humid summer, or warm summer and cool winter ("Warm" region); and the remaining seven areas have a hot dry summer, and mild or cold winter ("Dry" region; Fig. 2).

A total of 43,699 cases were reported in Queensland, Australia from 2001 to 2020. Among these cases, 13,422 were recorded in the Hot region, 25,949 were recorded in the Warm region, and 4,325 were reported in the Dry region. The mean incidence rate was five per 10,000 people per annum during the study period. Some SA3 areas on the coast and in Western Queensland reported high cumulative incidence rates across the 20-year study period (Fig. 2). The distribution of weekly RRV notifications in Queensland is shown in Fig. S1.

Ross River virus cases had strong seasonal variations with annual peaks in late summer and autumn (Fig. 3). Since 2006, RRV cases have increasingly been reported throughout the year. The highest notification number and highest incidence rate were recorded in 2015, while the lowest were reported in 2002. The weekly average values for maximum temperature, rainfall, and relative humidity at minimum temperatures are displayed in Fig. 3.

For each training dataset in each region, predictors (exposures at different lags) were selected to fit the standard negative binomial generalised linear model and then used in the training and validation of all the other models using the same dataset of this region. The coefficients and 95% confidence intervals of chosen predictors in the NB model using
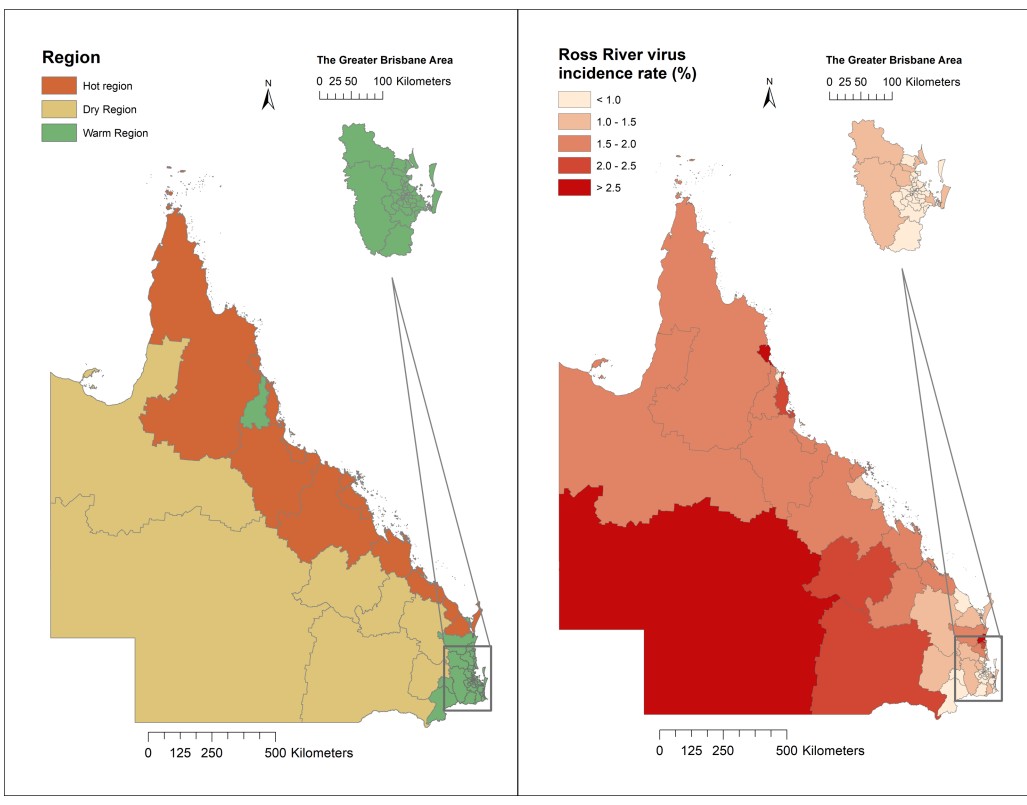

**Figure 2** Classification of regions and distribution of Ross River virus cumulative incidence in Queensland, Australia, 2001–2020.

the third training dataset for three regions are listed in Table 1. For each training dataset, nine to thirteen predictors were selected. Disparity in the predictor sets selected for the three regions indicates that regional analysis is necessary for detecting important predictors influencing RRV transmission under different environmental and weather conditions. The performance of models in preliminary analyses, which used the same model in selecting predictors and building models, are provided in Table S1. The results show that the NB model is appropriate for variable selection.

Evapotranspiration, NDVI, and vapour pressure with appropriate lags are selected for modelling RRV incidence in all regions, and a lagged El Niño variable was selected in three regions. Different lags for a single exposure may have both positive and negative effects on RRV incidence. In the Dry region, maximum temperature, vapour pressure and the index of education and occupation had positive effects on disease transmission, while evapotranspiration, El Niño events, proportion of agriculture land and NDVI were negatively related to RRV incidence. In the Hot region, rainfall, vapour pressure, evaporation, evapotranspiration, elevation and NDVI were positively correlated with RRV incidence. Temperature and types of land use were negatively associated with disease activity in the Hot region. In the Warm region, high values of weather exposures (relative humidity, vapour pressure and evapotranspiration) accelerated RRV transmission, while El

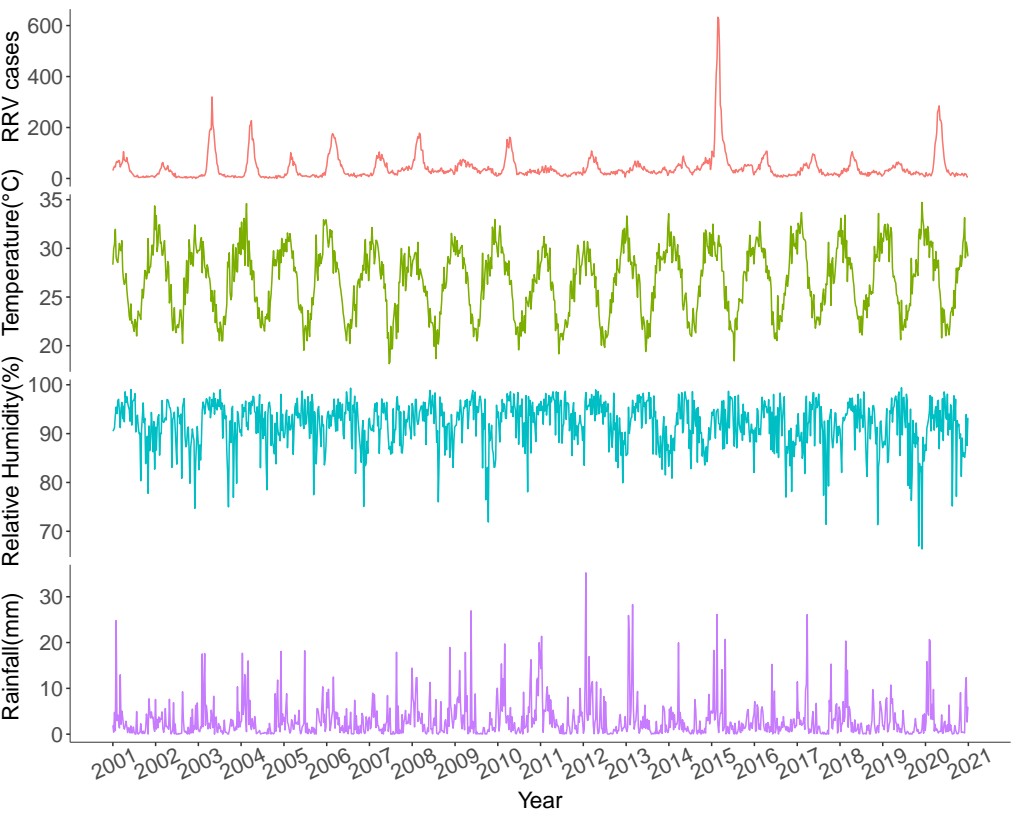

**Figure 3** Temporal trends of weekly total Ross River virus cases, weekly average maximum temperature, weekly average relative humidity at minimum temperatures and weekly average rainfall in Queensland, Australia, 2001–2020.

Niño events and high values of SOI and NDVI decreased RRV incidence. A high proportion of nature land and scores indicating less access to service centres also indicated areas with high disease activity.

The average AIC, BIC, and HQIC of the training datasets were calculated for each region and the whole of Queensland. The NB-based models had lower BICs than the Poisson-based counterparts, which indicated that over-dispersion exists in all datasets (Fig. 4). A lower variation in BICs indicated NB-based models were more stable than Poisson-based models. All NB-based models had similar BIC values regardless of model complexity across all regions. So, in general, NB-based models are robust, stable, and performed well in the epidemiological modelling of RRV surveillance data. The AIC and HQIC of all models are in Figs. S2–S3.

Within the Poisson-based models, Zero-Inflated models (ZI models) fitted the data better than the others, suggesting that the inclusion of zero-inflation in the model is needed if data dispersion is not considered. Considering the linear and non-linear models have similar BIC values, there is no evidence that non-linear models fit the data substantially better. Over the three model components considered here, the over-dispersion effect is

**Table 1  Coefficients of selected predictors by standard negative binomial generalised linear model in each region and in Queensland.**

| Region | Predictors selected by standard negative binomial generalised linear model[a] | Coefficient (95% CI)[b] |
| --- | --- | --- |
| Dry | Maximum temperature | 0.030 (0.014, 0.045)*** |
| | RHmin | 0.004 (−0.003, 0.010) |
| | RHmin, Lag 19 | −0.002 (−0.006, 0.002) |
| | VP at 3 pm | 0.029 (0.014, 0.044)*** |
| | VP at 3 pm, Lag 6 | 0.080 (0.068, 0.093)*** |
| | Evapotranspiration, Lag 37 | −0.095 (−0.150, −0.040)*** |
| | El Niño events, Lag 51 | −0.471 (−0.589, −0.353)*** |
| | Proportion of agriculture land | −0.013 (−0.018, −0.008)*** |
| | NDVI, Lag 14 | −3.960 (−5.259, −2.661)*** |
| | Index of education and occupation | 0.024 (0.020, 0.027)*** |
| | No El Niño events, Lag 51 | Ref. |
| Hot | Maximum temperature, Lag 37 | −0.059 (−0.069, −0.049)*** |
| | Rainfall, Lag 6 | 0.010 (0.007, 0.013)*** |
| | VP at 9 am, Lag 8 | 0.044 (0.036, 0.052)*** |
| | Pan Evaporation, Lag 19 | 0.146 (0.125, 0.167)*** |
| | Evapotranspiration | 0.157 (0.136, 0.178)*** |
| | Elevation | 0.001 (0.001, 0.001)*** |
| | Proportion of agriculture land | −0.028 (−0.033, −0.024)*** |
| | Proportion of water land | −0.241 (−0.265, −0.216)*** |
| | NDVI, Lag 48 | 1.156 (0.609, 1.703)*** |
| Warm | RHmin | 0.016 (0.012, 0.020)*** |
| | RHmin, Lag 2 | 0.016 (0.012, 0.020)*** |
| | RHmax, Lag 7 | 0.019 (0.016, 0.021)*** |
| | VP at 9 am, Lag 8 | 0.067 (0.061, 0.074)*** |
| | VP at 3 pm, Lag 11 | 0.061 (0.054, 0.068)*** |
| | Evapotranspiration | 0.148 (0.135, 0.162)*** |
| | Southern Oscillation Index, Lag 12 | −0.019 (−0.021, −0.017)*** |
| | El Niño events, Lag 6 | −0.569 (−0.622, −0.516)*** |
| | Proportion of nature land | 0.013 (0.011, 0.016)*** |
| | Proportion of tidal wetland | −0.05 (−0.058, −0.041)*** |
| | NDVI, Lag 15 | −2.824 (−3.137, −2.511)*** |
| | ARIA—Accessible | 0.498 (0.452, 0.544)*** |
| | ARIA—Moderately accessible | 0.136 (0.006, 0.267)* |
| | Number of weeks | 0.0005 (0.0004, 0.0006)*** |
| | ARIA—Highly accessible | Ref. |
| | No El Niño events, Lag 7 | Ref. |

**Notes.**
[a]RHmax, Relative humidity at the time of maximum temperature; RHmin, Relative humidity at the time of minimum temperature; VP, Vapour Pressure; NDVI, Normalized Difference Vegetation Index; ARIA, Accessibility/Remoteness Index of Australia.
[b]Significance: $p$ valu $e < 0.05$*; $p$ value $< 0.01$**; $p$ value $< 0.001$***
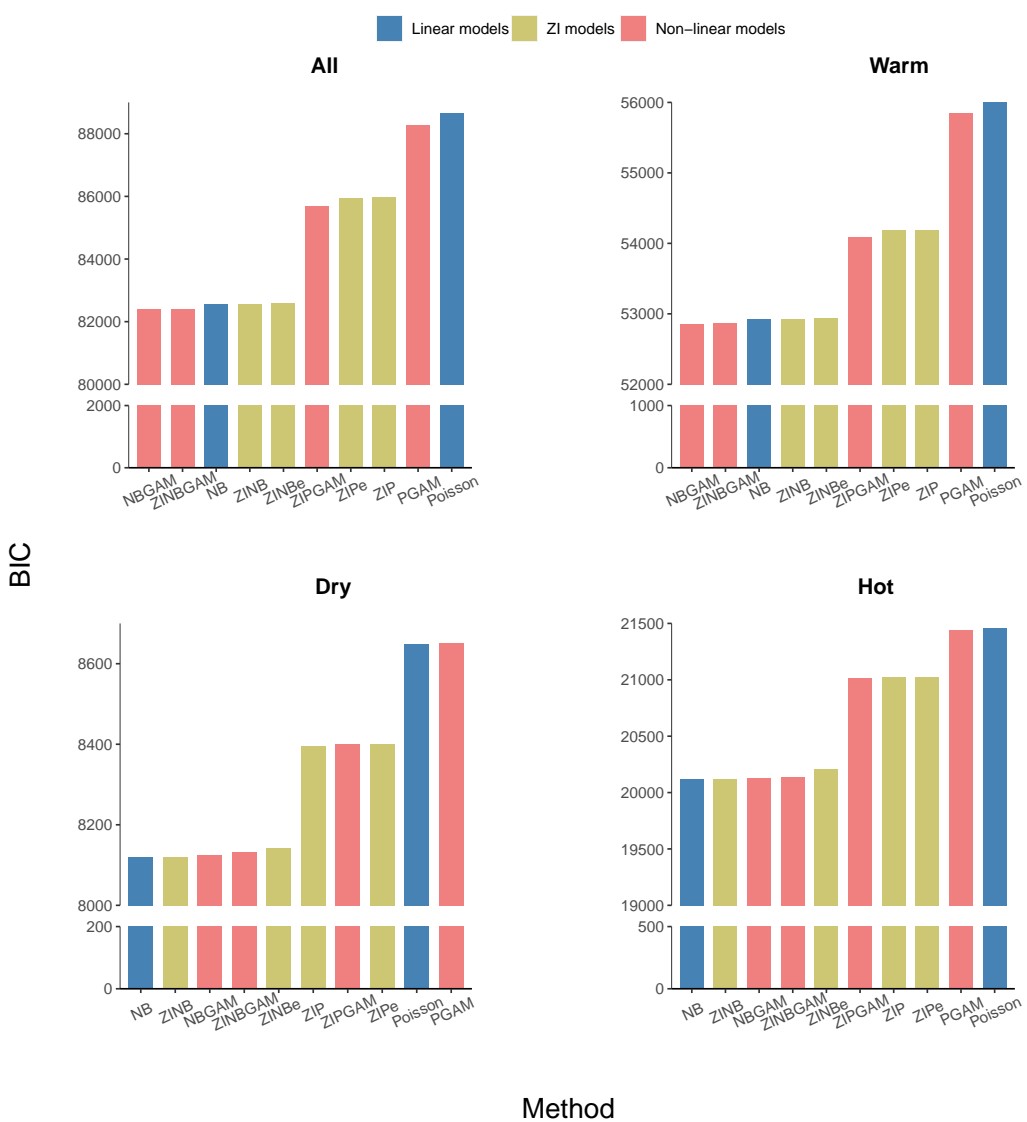

**Figure 4** Average BIC values of all models in different regions of Queensland (ordered by BIC value, with *y*-axis breakpoints).

most important, followed by zero inflation, with non-linearity being the least important in fitting these data.

The MSE, RMSE and MAE of the validation data in all regions are provided in Fig. S7 and Table S2 to show the unpenalized relative residuals of the models. These relative residuals were lower for the generalised linear models and ZI models. The GAM models with zero-inflation tended to have the greatest relative residuals in the validation sets.

The predicted values of the NB and the NBGAM models, which were stable and had lower BICs in general, together with real RRV cases in Queensland are displayed in Fig. 5. The results for the three regions are provided in Figs. S4–S6. These two methods were considered as the representative linear and non-linear models for this study. Both models

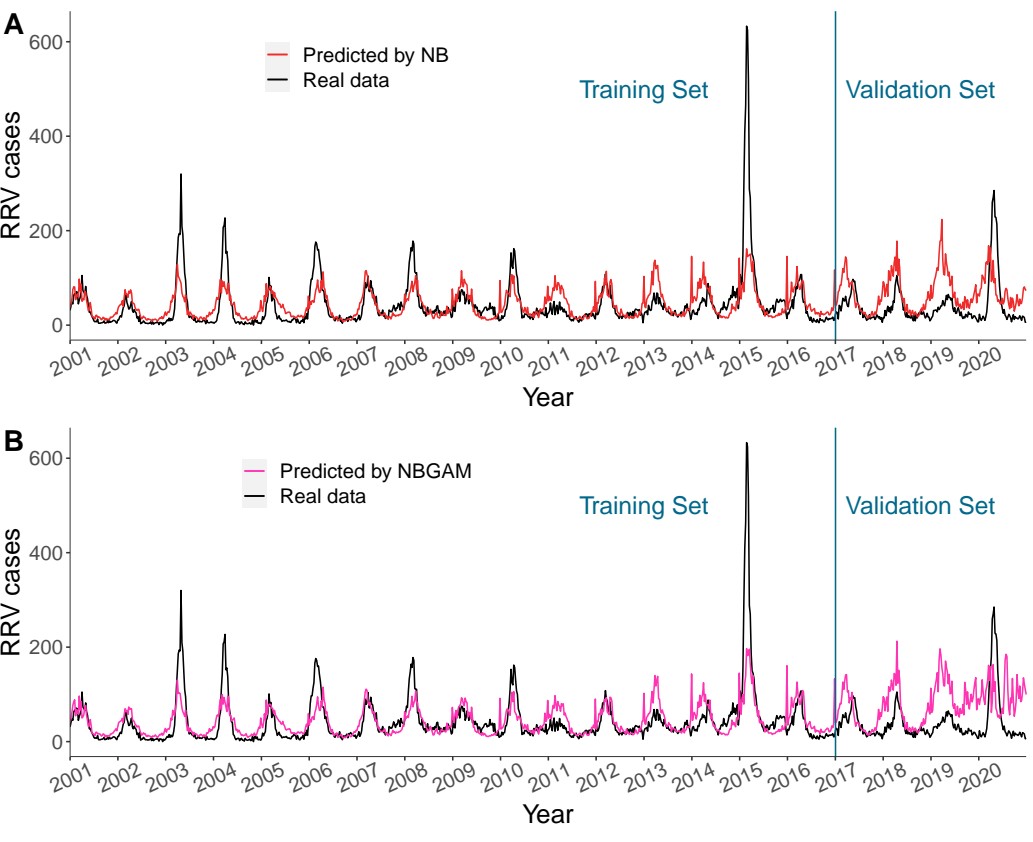

**Figure 5** (A–B) **Predicted value and actual value of NB model and NBGAM model for predicting RRV disease in Queensland.**

fitted the seasonal patterns and timing of peaks well but were unable to fit the size of the peaks, especially when the number of cases were high. Both models predicted one to two years of data comparatively well but failed to predict the latter two years. The non-linear models were less stable than linear models in prediction, especially in the final year, and over-fitting may have occurred.

## DISCUSSION

We modelled RRV incidence using climatic, geographical, and socio-economic exposures in different regions of Queensland. Using a substantial amount of data in a wide geographical area across 20 years with numerous exposures provided a reliable basis for model comparison. A purposeful selection strategy was developed to identify important predictors among numerous lagged variables. Ten models including generalised linear models (Poisson and negative binomial models), zero-inflated models (ZIP, ZINB, ZIPe, and ZINBe), and GAM models (PGAM, NBGAM, ZIPGAM, ZINBGAM) were implemented, where ZI models were applied to RRV data for the first time. We first compared model performance and evaluated the effects of over-dispersion, zero-inflation, and non-linearity of the epidemiological surveillance data on model fit. The standard

negative binomial generalised linear model fit the data well, and it is simple, robust, and efficient. Over-dispersion is the most important effect to be considered when analysing this type of data.

Most researchers applied a stepwise method for variable selection in developing RRV prediction models (*Koolhof et al., 2021*; *Gatton, Kay & Ryan, 2005*; *Bi et al., 2009*; *Ng et al., 2014*). Stepwise selection, an automated approach using either predictor significance or overall fit, does not account for a complex setting involving the modelling using correlated exposures at various lags. Limitations of stepwise selection include parameter estimation bias, inconsistencies among model selection algorithms, and an inappropriate focus or reliance on a single best model (*Whittingham et al., 2006*). Appropriate selection methods for climatic, environmental, and socio-economic predictors with lagged effects, taking account of complex correlations among predictors, have not been well studied. To choose important predictors, our strategy screens exposures at different lags based on both univariable and multivariable (adjusted) associations (the repeated backward stepwise screen and reassessment process). The significance and standardised coefficients of the predictors, correlations between predictors, and model fit are considered to select predictors. The stepwise process followed by a reassessment could avoid removing predictors based on a single criterion. The variable selection approach was able to identify the most important predictors by considering statistical significance, standardised regression coefficients, multicollinearity, overall fit and correlation between predictors in a comprehensive way. It was found to be effective and able to generate consistent and reasonable numbers of predictors across all datasets in this study, and we believe it has great potential in many epidemiological predictive studies including surveillance data arising from ecological designs. Different predictors were selected for each region and their effects on RRV activity varied across regions. Regional analysis is beneficial for selecting important predictors relevant to the transmission cycle in each region. However, adequate sample size is required to ensure a robust model fit.

Several exposures are proved to play relatively important roles in predicting RRV incidences, such as evapotranspiration, Normalized Difference Vegetation Index (NDVI), vapour pressure, and El Niño events. We found vapour pressure has a positive association with RRV transmission in all regions regardless of lags, as demonstrated in other studies (*Cutcher et al., 2017*). Evapotranspiration has a positive effect on RRV incidence in the Hot and Warm regions and a negative effect in the Dry region. As mentioned by *Koolhof et al. (2020)*, evapotranspiration may be a stronger predictor than other climatic factors including rainfall and temperature. The impact of El Niño events on RRV incidence has been less studied. We found a negative association between El Niño events and RRV incidence in the Warm and Dry regions. The decreased rainfall and higher temperature during El Niño periods may lower RRV incidence (*Australian Government, 2020*). Vegetation cover favours mosquito breeding and provide habitats for reservoir hosts. However, in forests with dense vegetation, mosquitoes may be less likely to survive and breed because of the low temperatures in the shade. The average NDVI value is high after rainy days in late autumn and early winter in Queensland when the disease incidence is low (*Australian Government, 2019b*). These might partly explain our findings that high NDVI at three

months' lag in Dry and Warm areas was associated with higher RRV activity, while high NDVI in the Hot regions around 10 months earlier may lead to vegetation growth in forests and cause decreased RRV incidence (*Ha et al., 2021*). A high air temperature may prevent mosquito breeding, thus may decrease disease transmission in the Hot region, whereas high temperatures in the Dry region may help mosquitoes to survive through a cold winter and subsequently increase RRV incidence (*Jacups, Whelan & Harley, 2011*; *Werner et al., 2012*). Relative humidity lagged at 1–2 months may accelerate mosquito breeding and vegetation growth, thus indirectly contribute to disease transmission (*Ng et al., 2014*; *Tong & Hu, 2002*). The selected predictors, such as vapour pressure, NDVI, and El Niño events, can provide information about conditions that could accelerate mosquito breeding or indicate high-risk areas that require disease prevention. Targeted mosquito surveillance and control programs (*e.g.*, mosquito monitoring, reduction of breeding sites, and control of larval and adult mosquitoes) can reduce transmission of the disease. Interaction between exposures was not considered here due to the large number of exposures; however, there is potential to develop models that capture all three regions simultaneously, with interaction terms used to capture differences between regions.

Modelling RRV is challenging because of the complicated transmission cycle, the characteristics of aggregated notification data, the wide geographical area of Queensland, and the 20-years of data. To avoid the limitations of a single method, we considered the effect of over-dispersion, zero-inflation, and non-linearity, and added corresponding components into the models. Over-dispersion of the data may be due to excess zeros, due to the distribution of RRV cases in SA3 areas, and models that incorporate consideration of over-dispersion were found to fit the data better. Indeed, we found that the standard negative binomial generalised linear model fit the data well suggesting that it's ability to model over-dispersion may have been sufficient to solve to problems of excess zeros, which often manifests as over-dispersion. Non-linearity can be a good addition to NB models, but the GAM models did little to improve model fit without considering dispersion or zero-inflation of the data. Generalised linear models and ZI models performed better than GAM models, with or without zero-inflation, based on relative residuals in the validation sets. Our aim is to build and obtain models with excellent predictive performance and clear epidemiological logic in different regions, which is to balance a good model fit in the training set with selected explainable exposures and a good model predictive performance in the validation set. Generally, among the models implemented in this study, NB-based models were the best choice and are recommended for similar epidemiological studies that consider vector-borne diseases at different temporal and spatial resolutions.

Two noteworthy peaks of RRV notifications were reported in 2015 and in 2020. The 2015 outbreak were possibly due to a combination of ecological factors including above-average rainfall and consequent high mosquito abundance (*Jansen et al., 2019*). The high cases reported in 2020 may be related to altered RRV transmission risk and reporting. Changed behaviour due to COVID-19 may have resulted in more time gardening or exercising outdoors, while awareness of symptoms of fever and fatigue may have affected healthcare seeking (*Webb, 2020*; *Jansen et al., 2021*). The 2015 outbreak was included in our training set and the 2020 peak was in the validation set, allowing a test of model performance when
RRV incidence was extremely high. The predictors selected could explain the common trends of RRV incidence but were unable to predict the rise of disease transmission in 2015 and in 2020. Historical RRV cases, modelled with an autoregressive term, may help to detect incidence within a certain range and over a short period. Some important exposures, such as mosquito abundance and host populations, have been suggested as greatly improving model fit (*Ng et al., 2014*), but are not included in this study as they are not available across Queensland for our study period. Adding autoregressive terms and using different formats of variables (*e.g.*, using the number of rainy days instead of average rainfall in a week) may improve model fit and predictive performance.

Data quality has an impact on modelling performance. The definition of RRV was revised in 2013 and 2016 to remove possible false-positive diagnostic test results (*Selvey et al., 2014*; *Knope et al., 2019*). We are unable to evaluate the impact of these revisions on predicting RRV incidence. As a passive surveillance system, RRV surveillance may be biased for the following reasons: underestimation caused by asymptomatic infections, false-positive diagnostic tests (*Selvey et al., 2014*), greater efforts consequent upon greater awareness by doctors to ascertain cases during outbreak years (*Werner et al., 2012*), and lags in reporting due to the incubation period (*Queensland Government, 2017*). Flooding and bushfires were coded as binary in each SA3 area and do not capture areas influenced by these events or the river height of related river catchments. The SEIFA and ARIA data from censuses in 2011 and 2016 had to be extrapolated to the entire study period. Data on land use and elevation were only available in one year, thus were considered as constant values in this study, with spatial estimates of elevation interpolated to areas with missing data. The weekly population data required linear interpolation and extrapolation when converted from yearly data. Data available with adequate quality were considered and data were processed with reliable procedures.

Potential methodological limitations might exist when we determined data units for analysis and chose models for variable selection. The weekly SA3 area data used in this study contain details but have many zeros. However, using data at a lower spatial or temporal resolution (*e.g.*, monthly data) have fewer zeros but may lose information, especially for daily weather data. The standard negative binomial generalised linear model was used for selecting predictors for all models which may have given it an advantage in model fitting. However, using the same set of variables makes head-to-head model comparison possible. We chose here to select the same set of predictors for all models to enable a comparison of different models, rather than a comparison of the model building process, with our additional analyses suggesting that any advantage to the negative binomial model is relatively small.

## CONCLUSIONS

This is the first head-to-head comparative modelling study of prediction of RRV incidence in Queensland. We developed a comprehensive strategy to purposefully select variables using multiple criteria followed by reassessments and to allow the choice of a stable and valuable variable set for model fit. The comparison among different types of models

provides an example of determining the key element affecting model fit. We found the standard negative binomial generalised linear model to be the best choice in predicting RRV incidence and the statistical over-dispersion effect possibly caused by excess zeros is a principal issue to be considered. We advocate a model selection algorithm, such as what we developed, that accounts for some issues arising while modelling infectious disease incidence including various exposures, lags, data distribution, multicollinearity, and model fit. The model building process introduced is promising for epidemiological surveillance studies of mosquito-borne disease with temporal and spatial data.

## ACKNOWLEDGEMENTS

We are grateful to Frances Birrell, the epidemiologist for the Epidemiology and Research Unit, Communicable Diseases Branch, Prevention Division, Queensland Department of Health, and Mayet Jayloni, the senior data manager for the same unit, who supported us greatly in obtaining Ross River virus notification data. We are thankful to Mr. Ben Hague from Climate Data Services, Bureau of Meteorology for providing help in collecting climatic data. We appreciate the great advice from Dr. Amanda Murphy at QIMR Berghofer Medical Research Institute and Dr. Eloise Skinner at Stanford University. We are grateful to Patricia Dale, Emeritus Professor at Griffith University, Michelle Gatton, Professor at the Faculty of Health, School of Public Health and Social Work, Queensland University of Technology, and A/Professor Paul Dawson, Honorary Associate Professor at the Mater Research Institute, University of Queensland, for their professional comments and suggestions. Australian governments fund Australian Red Cross Lifeblood, where Dr. Viennet is employed, to provide blood, blood products, and services to the Australian community.

### Funding

This work was supported by the University of Queensland Research Training Scholarship and Frank Clair Scholarship. The funders had no role in study design, data collection and analysis, decision to publish, or preparation of the manuscript.

### Grant Disclosures

The following grant information was disclosed by the authors:
University of Queensland Research Training Scholarship and Frank Clair Scholarship.

### Competing Interests

The authors declare there are no competing interests.

### Author Contributions

- Wei Qian conceived and designed the experiments, performed the experiments, analyzed the data, prepared figures and/or tables, authored or reviewed drafts of the article, funding acquisition, and approved the final draft.
- David Harley conceived and designed the experiments, authored or reviewed drafts of the article, funding acquisition; Supervision, and approved the final draft.

- Kathryn Glass conceived and designed the experiments, authored or reviewed drafts of the article, supervision, and approved the final draft.
- Elvina Viennet conceived and designed the experiments, authored or reviewed drafts of the article, funding acquisition; Supervision, and approved the final draft.
- Cameron Hurst conceived and designed the experiments, performed the experiments, analyzed the data, prepared figures and/or tables, authored or reviewed drafts of the article, supervision, and approved the final draft.

### Human Ethics

The following information was supplied relating to ethical approvals (i.e., approving body and any reference numbers):

The University of Queensland Human Research Ethics Committee A granted Ethical approval to carry out the study within its facilities (No. 2019/HE002772).

### Data Availability

The collated weekly data and the R code is available in the Supplementary Files.

Direct access to the raw data requires approval from the Communicable Disease Branch (CDB), Queensland Department of Health (https://www.health.qld.gov.au/clinical-practice/guidelines-procedures/diseases-infection/surveillance/reports/notifiable/data-request, or email EPI@health.qld.gov.au).

### Supplemental Information

Supplemental information for this article can be found online at http://dx.doi.org/10.7717/peerj.14213#supplemental-information.

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
