# Peer review of "Prediction of Ross River virus incidence in Queensland, Australia: building and comparing models"

_PeerJ, doi:10.7717/peerj.14213_

## Round 0.1 · original submission · Major Revisions

Two reviewers have commented on your work. I feel all of their comments warrant a response on your part and will invite you to prepare a revised version of your manuscript, along with a point-by-point rebuttal of the reviewers’ (and my) comments. I should say that there is much that I like about your manuscript and I am looking forward to seeing the revised version and your responses.

Reviewer #1 is very positive and raises only a few points for you to address, including around the content of the background, some possible references for you to consider (note that you are not obliged to use the reviewer’s suggested references and may decline their suggestion here or prefer other references instead), and the implications of your study.

Reviewer #2 has made a very thorough and thoughtful review of your work and raised a number of significant issues, alongside some smaller ones. I agree with them that having the data available would be preferable, but I appreciate the limited options that I imagine are available to you at the present time. Would it be possible to make the data available for review purposes only? Or to provide simulated data that is broadly consistent with the actual data? Their point about how much this is intended to be a methods paper is important to consider. I have sympathy with their point that considering other methods would be useful in that context and would potentially make the comparisons more realistic in terms of candidate methods (although you could also argue that these methods already cover what you would anticipate models to be based on). Their point that the negative binomial regression model was used for variable selection seems especially important to me. I would expect this to give a, perhaps non-trivial, advantage to this method (e.g., a variable associated with zero inflation could be omitted here as could a variable with a non-linear association with the counts) and could not follow the reasoning here (or Lines 331–333, where I would expect this would definitely provide an advantage of some size to this method and would only seem to produce a fair head-to-head comparison once the set of variables is determined). I would have expected that a real-world comparison of methods would involve variable selection using the method being examined at that time (c.f. Line 71).

While Hosmer-Lemeshow style approaches to variable selection remains popular, as do stepwise and IC-based approaches, as an empirically-driven variable selection strategy, this would be expected to produce estimates too far from the null and p-values too low when used on a single data set. For exploratory model development, this is less important; for confirmatory research, these are clearly problematic. I think it would be helpful to explain around Line 129 why you’re using this strategy (e.g., because you see the choice of methods as related to exploratory research, because you are using the approach that you feel other modellers would also use, or some other reason). Related to the variable selection approach mentioned above, I feel that the process should as closely as possible reflect that used by a competent researcher when developing a model. This might simply require a sentence noting that you consider your choices to represent common practice (c.f. Line 43 onwards) or at least reflects statistically acceptable practices that seem likely candidates for future research in this area. I wondered why hurdle models didn’t feature (c.f. Line 56) in your candidates, although I appreciate that your ten methods could easily become twenty or thirty following this and related arguments.

A similar point would apply for me to the check for multicollinearity on Lines 139–140. Correlations between exposures won’t bias your results, so I think you should explain the reason you are doing this here (beyond redundancy, unless data collection is a factor) and provide a reference or other justification for using Rho > 0.9 as the criterion. Similarly, a reference for using VIF > 10 as a criterion for removing variables (Line 145) would be helpful for readers as would explaining the “problems” on Line 143. It was initially unclear to me why the VIF > 10 removal criterion (Line 145) was followed by a VIF > 5 criterion (Line 147). In fact, in Figure 1, it became apparent that there was a three step process for VIF-based removal (VIF > 5, VIF > 10, and then VIF 5 again) and this then explains the BIC-based removal criteria which seemed absent from the methods text. I’m not sure that a reader would be able to write a version of the statistical methods from Figure 1 alone, or construct Figure 1 solely from the statistical methods and wonder if the two could be more closely reconciled here. I appreciate that S1 contains more information.

Related to the above, some of the material in S1 seemed quite close to what was already in the manuscript and at least some of the rest added what could be seen as important information (e.g. about the VIF and BIC-based variable selection). I wondered if some of this material, there is not a lot there, could be moved into the body of the manuscript. I was less enthusiastic about my suggestion here for the final section on model performance, but will note that even some of the equations (particularly 1 and 2, 3 and 4 could perhaps be described in general terms for most readers, and then 5) for the models being considered could be helpful to researchers who need a reminder about or introduction to these.

Line 137: Do you mean “univariable” here (i.e., with a single exposure variable; rather than “univariate”, i.e. with a single outcome variable)? See also Figure 1 and possibly elsewhere.

Reviewer 1 ·

Basic reporting

1. Clear and unambiguous, professional English used throughout. Yes, the manuscript is well written and presented. The methodology, results, and discussion clear.
2. Literature references, sufficient field background/context provided. Generally yes. I feel it is always a difficult balance when presenting papers of this nature with regard to providing enough background on the general approach of modelling mosquito-borne disease and providing sufficient information on the ecological/biological background on mosquitoes, their environmental associations, and matters related to arbovirus transmission cycles between vectors, reservoir hosts, and humans. Personally, I would have preferred to see more information on these matters provided with examples of the ecological factors influencing vectors in the three different regions highlighted in this study. In the discussion, the authors themselves state that "Regional analysis is beneficial for selecting important predictors relevant to the transmission cycle in each region" and some more detail here to "set the scene" may have been useful. I appreciate that this information is generally readily available from a number of previously published paper so may not be critical here.
3. Professional article structure, figures, tables. Raw data shared. Yes. Manuscript has been well structured and professionally presented.
4. Self-contained with relevant results to hypotheses. yes.

Experimental design

1. Original primary research within Aims and Scope of the journal. Yes.
2. Research question well defined, relevant & meaningful. It is stated how research fills an identified knowledge gap. Yes.
3. Rigorous investigation performed to a high technical & ethical standard. Yes.
4. Methods described with sufficient detail & information to replicate. Yes.

General comments. I should note that I do not have extensive experience in the field of modelling. However, I am familiar with the body of work undertaken in this space and I am satisfied that the approaches presented here are consistent with the scientific rigor of existing literature. I do not have any specific comments on the experimental design presented here.

Validity of the findings

I am satisfied that the outcomes of this investigation are scientifically sound. My only general comment is that the authors may want to specifically address two noteworthy peaks on number of cases of RRV during 20156 and 2020 and how these exceptional events may influence analysis specifically. Perhaps a brief comment is required with respect to reports about teh 2015 outbreak (as provided in Jansen, C.C., Shivas, M.A., May, F.J., Pyke, A.T., Onn, M.B., Lodo, K., Hall-Mendelin, S., McMahon, J.L., Montgomery, B.L., Darbro, J.M. and Doggett, S.L., 2019. Epidemiologic, entomologic, and virologic factors of the 2014–15 Ross River virus outbreak, Queensland, Australia. Emerging infectious diseases, 25(12), p.2243.) and the unusual circumstances of the 2020 outbreak coinciding with COVID-19 and associated social changes in human activity (reported in Jansen, C.C., Darbro, J.M., Birrell, F.A., Shivas, M.A. and van den Hurk, A.F., 2021. Impact of COVID-19 Mitigation Measures on Mosquito-Borne Diseases in 2020 in Queensland, Australia. Viruses, 13(6), p.1150. and Webb, C., 2020. Reflections on a highly unusual summer: Bushfires, COVID-19 and mosquito-borne disease in NSW, Australia. Public Health Research & Practice, 30(4).)

Additional comments

There has been considerable work done in the field of modelling RRV activity, as evident in the 43 studies identified here by authors. However, what is generally not addressed in the level of detail that is often required is how readily this work is translated for application by health authorities. Some considerations have been touched on by authors here but it would be useful to include a brief comment on how the outcomes of this work should be practicably applied to the management of mosquito-borne disease risk by health authorities in QLD, or elsewhere in Australia.

·

Basic reporting

I thank the authors for their hard work on a very important topic. Considerable effort has been made to follow best practices for the data preparation, cleaning, and organisation and deep thinking has gone into how to incorporate and discuss lagging of variables appropriate to this study system. Consideration research has also gone into the implementation of a variety of variable selection and modelling approaches that go considerably beyond what has been attempted in this literature. With that said, I have some major methodological questions I would need answered to see this paper published (see part III), and would also recommend a thorough revision and reconsideration of the framing of the aims and focus of the results/conclusions (see part II). I will summarise some of my most crucial comments in sections II and II where appropriate, but provide the rest in the order they appear in the manuscript under General Comments.

In terms of basic reporting, the language is well written. But I had to read it twice to answer several methodological questions I had upon first reading. These are detailed in the line by line below. For example, the question of which model was used in the variable selection, and details about the range of lags tested for each variable, among others. The introduction I feel is lacking some essential context on the performance of existing models, the gaps in that performance, the variables that have been found relevant in the past. It relies heavily on a previous review, which is likely an excellent source for readers to check, but some more of that needs to be brought to this paper to fill in the justification for this work. I have a lot of comments on Figure 1 below. I commend the authors for preparing such a “methods” figure, but it is very busy and quite confusing. I would also like to see substantial changes to Figure 4, which I find doesn’t not communicate very well. I would also love to see predictive performance on test data for the different models shown in the main paper, as well as the variables and their lags that were ultimately selected.

The data is listed as available upon request. I’m not sure how PeerJ handles potentially sensitive health data, but I imagine they are deidentified, and I at least would ask for the data to be provided for review, and the covariate data and a simulated or further deidentified subset of the case data provided for the reader.

Experimental design

This is in the scope of the journal. I think the work is very useful for the body of literature and managers attempting to forecast Ross River Virus incidence, but the explanation of this link is not very well supported in the introduction. In fact, connecting this work to forecasting RRV or better understanding predictors is almost absent. In addition, I find the aims unconvincing. From my reading, aims 1 and 2 were very similar, aim 3 was more of a step along the modelling process than a separate aim, and no aim was listed to address the interesting and vital components of the impact of these new models on our ability to forecast disease incidence or better understand environmental predictors of RRV and how they vary across space. This latter point I think the authors should examine more deeply, as their work has the possibility to address it in a very compelling way with the regional models. It seems the authors want to frame their paper as a methods paper, and although they do great work in this space, I do not think that should be framed as its main purpose. A true comparative modelling paper I imagine would have to incorporate some non-regression approaches such as BRTs or random forest. In addition, and perhaps more importantly, I think there is an opportunity to spend more focus on the forecasting/predicting implications, what we now know about impactful variables/relevant lags. If these models improve on those currently in use by QLD, it represents a big opportunity for better public health managing, and this is important.

Validity of the findings

I have a few important questions about the findings/the way they are reported. I did not see anything in the aims or methods that related to the model results shown for separate climatic regions. I do not know where those region designations came from, whether using them was part of the original intent or they were added later to improve model performance, and little is done to elaborate or query what predictors are predictive in the separate regions and why. I do not recall seeing a breakdown of case reports by region, which could be highly imbalanced. I would have loved to see a version of the model for all the data that incorporated region as a categorical variable, or even in an interaction with other environmental predictors. This may be more appropriate for a follow-up work if this is too much for the current manuscript, but discussion of this as a potential approach for future implementation would be perhaps a nice addition. There seems to be a lot more to tease out from the regional differences. In addition, a major conclusion of the work is that the negative binomial performed the best, but this model form was used in the variable selection step. I’d love to see the authors test out the impact of variable selection using the standard Poisson model, and check if the results differ.

Again data have not been provided so I was unable to check them.

Additional comments

Abstract
• 31-33 “Compared to non-linearity of predictors and excess zeros, the over-dispersion of the data is the primary factor for model fit.” This sentence wasn’t clear to me when I read it but became clear after reading the rest of the paper. I would consider rewording.

Intro
• 49 What vector and reservoir host species? This is interesting and important! Coming back on second read to ask - How did they differ across the regions you look at later? Do the different vectors help explain the different selected best predictor variables?
• 52 Sounds like it is summarising the results of another paper. Would be useful to reframe this in a way more typical for an introduction. What is the gap in implementation approach you are filling and why? In general I feel the introduction is too brief and I would want to see more detail on the epidemiological justification for various predictors/lags as related to indirect modelling of vector species, how modelling approaches would improve change management/reporting, what is missing in our current understanding/implementation. RRV modelling is hard! So I think placing more details here will not only help readers who are working on other diseases relate to the issues you are trying to solve but will help folks modelling RRV think about what issues they may not have noticed in their own models/approaches.
• 61 I find this approach paragraph confusing. I would start with your aim and follow with the approach, or integrate the two (“we used XX models to address XX aim”). How does your aim build on the gaps you identified in the previous paragraph summarising what has been done?
• 70 Aim 2 sounds a lot like aim 1… where is your aim looking at the effect of various predictors on RRV incidence? What about improving performance of RRV models for predictive performance into the future? Anything about the regional distinction you employ later in the paper?
• 71 I find Aim 3 a bit surprising. You demonstrate a strategy for variable selection, but it doesn’t seem like an aim of the paper. Perhaps if you were testing and comparing quantitatively several variable selection approaches, then it would be an appropriate aim.

Methods
• 131 – 134 I like the distinction and explanation of “exposure” vs. “predictor” where predictor is exposure + lag
• 137 Univariate analyses of individual predictors against RRV cases > Sounds like you calculated correlations of lagged exposures against RRV cases to rule out/whittle down the lagged exposure variables to include the rest of the selection steps. Which underlying model? Which lags? Weekly from lag 0 to lag of 1 year? Monthly? You mention only up to 1 year, but at which interval before a year? Some of these questions are answered later but they beg to be answered here. (also address this in S1).
• 145 it is not clear to me why the variable selection step was conducted separately for climatic then geographical/socio-economic variables, then together
• 150 it says “We built models for longitudinal data involving exposures at different lags” I imagine from the previous section that this refers to models using the variables previously selected, of which there will be lags of different lengths for different exposures. But it is a little confusing, and sounds a bit like building different model options with multiple possible lags for the same exposure variable…
• 156 Excellent. Yes a time-series cross-validation is what is needed in this instance.
• 157 However I am concerned about the differing size of the training blocks between the three folds. The fold trained on more data will always perform better than the fold trained on less data. I suppose if this bias was propragated for each model type compared, then perhaps it is not an issue. In Figure 4 when you show average BIC for each method for each region, that is the average across these three folds? I think these should be boxplots or another device to more easily visualise the variation/spread. Also would help with tall y axis that makes the variation between models hard to discern.
• 162 Sentence beginning with “Non-linearity…” seems like the wording is off, does not read like a complete sentence.
• 160, 163 – 166, 167 You start by saying Poisson and NB are appropriate for this data. Then say that you add NB, zero-inflated, and non-linear, in contrast to the standard or baseline Poisson model. Then you say you used NB as the baseline model for the variable selection. I think you should sharpen and clarify this language to be more clear about your process and the justification for it. The three dimensions of variation between your selected models as shown in Figure 1 is informative, but does not provide a good explanation for why NB was chosen for the variable selection step. And the way it is described here sounds like these three dimensions are in contrast to the standard Poisson, which perhaps is an unnecessary division. You can perhaps just highlight the variation in the three dimensions and that the models selected allow you to evaluate the fit in these varied ways?
• 176 why were the models fit in each region separately, before all together? Were there region specific effects that were expected, that you were not able to fit in a hierarchical structure by region? What was your expectation that lead to this design? Revisiting this comment about reading the whole paper, I see the answers to some of these now, but they beg an explanation here.
• 186 In the beginning of this paragraph you discuss model fit of the training data, but here I think you should replace “model fit’ with “model predictive performance” to distinguish the metrics used on the validation data.
• 191 – 193 Thank you for citing your R packages

S1
• Very nice to have this metadata of the variables
• Data collation steps 1:3 all seem to be about downloading the data. And many of the other steps seem more like code instructions than meaningful variable processing records (eg. “4. Convert data from wide format to long format”). I think this table could be simplified to exclude some of these categories so the more crucial processing steps are clearer, (eg. 13:16)
• In Model building, GLM section:
o You state “log transmission of RRV incidence follows a Poisson distribution and can be predicted by a linear combination of independent exposures.” Should this be instead “transmission of RRV incidence follows a Poisson distribution and log transmission can be predicted by a linear combination of independent exposure”?
o Perhaps specify after the equation when you introduce g() for the link function that it is log link for both Poisson and NB.
• In Model building, Zero-inflated models section:
o The final paragraph of this section starting with “Considering that weather exposures predict RRV incidence well and lags in three months are epidemiological reasonable,…” I find confusing. Earlier you describe lags up to 1 year. Here they are up to 3 months/13 weeks.
• In Model building, GAM section:
o It would be helpful for the reader to see the various models with consistent acronyms/labels. In figure 1, the four models discussed here are given other names/slight differences (eg PGAM v. Poisson GAM). Perhaps a table somewhere that introduces the full name, the acronym, the key distinguishing features along those three scales, and a citation?
o You state “complicated relationship between lagged exposures and RRV incidence”, where does this come from? Visual inspection? Other prelim analyses? References?

Figure 1
• I originally wondered “What does “Select predictors by NB model” mean, if the following steps describe variable selection?”, then realised the yellow squares were headers for the subsequent steps. And realised NB model was used in variable selection. This is not very clear. These headers do not need colours, and would perhaps be aided by numbers? So: “1. Split training sets… 2. Select predictors…” etc. Except then it reads like the predictors are selected after the data are split. Is that correct? If so did each training fold have different predictors?
• There’s a lot going on in this Figure, which I realise makes it difficult to design. But I think the authors should spend some more time on reducing the visual complexity. Some ideas:
o I’m not sure the three panels are needed. From what I can understand by the “Model building process” panel, splitting the sets happens first, then steps 1-5, and finally the model building. I think the figure could be rearranged to show those three stages one after the other.
o Perhaps Steps 1 – 5 do not need to be shown twice, once in long form and the other in short form.
o Remove the shading on the titles, it is distracting
o Consider reducing the colours. There are too many to be meaningful. The training vs. validation differentiation could be achieved with just the solid/dashed lines, and as I’ve said I don’t think the model process steps (in yellow) need to be shown that way.
o The 3D diagram for the model forms is compelling in that it shows how the 8 models differ from one another along three axes. But what does the red colour represent? The box lines do not appear straight/even, which is a bit disconcerting. I think you could simplify as well with simpler arrows. If it were me, I would find a way to rearrange so that the simplest model versions are at the top (‘standard poisson” and ‘standard negative binomial’ because readers are reading down the page and encountering “ZIP GAM” and “ZINB GAM” first. These will be obvious if they are guided from the standard models.

S2
I commend the authors on supplying their code. I have taken a look and I appreciate the header information, clustering of packages used at the top, and commenting. With that said, I can do very little to verify or check the code/results without access to the data. I understand that health data, even deidentified, may be restricted. But if it could at least be available for review, I would like to run the code and check your output. In addition, it seems part of the thrust of your paper is illustrating methods and approaches. In this case I think it would be great to provide *some* data to play with the code with, even if the actual RRV case data is, naturally, a bit sensitive. Could you perhaps provide the covariate data and a simulated version of case data with the paper? I leave it to the editor to assess whether this is sufficient for PeerJ policy.

Results
• 199 – 203 where do the region classifications come from? Are these from clustering your own data, or bioclimatic classifications from the State? A citation or justification would be useful here. I’m surprised these are coming up for the first time here, when they should have been introduced in methods.
• 207 I think this histogram shows distribution of weekly cases? Should add that clarification here and in S2.
• 213 “For each training dataset, predictors .. were selected” does this mean for each cross -valiation split? Or for each region model? Surely predictors were kept constant across folds for a given region?
• 218 – 219 the conclusion here is a bit strange to me. I think you’re saying that the environmental conditions in each region select for different most useful predictors, but this is predicated on the environmental variables that are used to define the regions themselves, which we have not been shown. I understand if the authors do not want to run additional models to test, but one recommendation for discussion will be thinking about how to incorporate the effects captured by the region designation in future models. For example with interaction terms of temp/precip in summer. Or even interaction with the region designation itself.
• Fig 4 – instead of colouring by model types (these are already separated in the four panels) perhaps you could colour the methods by the groupings you defined? That would allow us to visualise, where are the nonlinear models? Where are the zero-inflated? This is complicated because the types overlap, but somehow visualising this will be more informative for the reader. It’s very hard to compare the outputs in this style. I would also consider a boxplot approach, as its hard to compare the bars here.
• I would like to see the different predictors selected in each model, this is a key result.
• Fig 5 – I would prefer to see models presented here that demonstrate the best performing on validation data. I think for a disease such as RRV, forecasting is much more important than fit to training data.
• On that note, would love to see a figure of the absolute error of each model against the validation data.

Discussion
• 259 “most researchers” begs for more than one citation
• 267 what’s the multivariate step in the variable selection strategy?
• 272 I find the conclusion that the variable selection approach was “stable” a bit strange considering a different set of variables were chosen for the four different datasets compared. Do you mean when this is done for different subsets of the same data, the same predictors are chosen? If so, I didn’t see this result.
• 277 The variables selected and their lags should be introduced in the results
• 286 – 291 this discussion is interesting. Will reiterate that the variable selection results should be shown in the results to better lead the reader here.
• 289 I see that you mention the potential impact of using NB to do variable selection in 331 – 333, but I think that qualification should sit up here with this content, and you may need to probe it more deeply, eg by running them (or a subset of them) with a Poisson model.
• 301 – 305 Considering a main conclusion from this study is the model comparison step, I would encourage the authors to find other ways to visualise these comparisons. Looking back at the figures presented I would not have arrived to the discussion convinced of this conclusion. I think part of this is making a clearer distinction of the model fit v. predictive performance measures (ie metrics to evaluate fit of model to training data, v. predictive performance to the left out test data).
• 306 – 307 This appears an important conclusion. I would love the authors to expand on their explanation here. Also, many mosquito models themselves use the type of environmental variables you are employing, so I would imagine you are indirectly using predicted effect on mosquitos?
• 327 I appreciate the effort to list some possible limitations, but I find the conclusion at the end that they may have had “minor effects” a little unsatisfying. Is there any other work you can cite on the expected impact of some of these limitations?

---

## Round 0.2 · Minor Revisions

Thank you for your responses, to both my questions and those from our reviewers. I appreciate your constructive edits and additions, which I think will help readers to understand your interesting work. My remaining comments are very minor (see below).

I’m satisfied by your responses to most of my questions, including those about the data availability and variable selection. I think readers would appreciate a table note under Table S1 explaining the “Not studied” entries.

As an aside, I’m not entirely convinced by your response about multicollinearity (in your response to my point #5, “Multicollinearity may distort coefficient estimates of the model (e.g. turn possible risk factors into protective factors), and may change the statistical significance of the predictors.”) as I would see this as more an issue of interpreting any variable after conditioning on other variables in the model. Multicollinearity would increase the uncertainty around estimates (but by less than some of my collaborators seem to imagine), and so would increase the chances of nonsensical results, but with observational data, it’s rare that coefficients can be interpreted without awareness of all of the other variables in the model. However, I agree with your point about parsimony, and a goal of model stability is easy to justify here, so I’m satisfied with your response to this point also.

I’d perhaps quibble with demonstrations of external validity being entirely achievable using cross-validation (in your response to my point #3), but what you’ve done is clearly a pragmatic solution to a real-world problem, so I'm happy with this also.

My question about hurdle models (#4) might still be one that occurs to other readers (particularly given your Lines 62–62), so I’d like a short (one or two sentences?) explicit justification for their exclusion from the candidate models, perhaps somewhere in your methods.

Reviewer #2 has raised one substantial point about the selection of the negative binomial model which needs addressing (and I’d suggest tweaking the x-axis labels in Figure 4, and also Figures 3, 5, S2, S3, S4, S5, S6, and S7, so that they are not overlapping with the axis itself). They’ve also made a few other smaller comments that are worth acting on or considering.

·

Basic reporting

I would like to thank the authors for their thoughtful consideration of the comments I made in the first round.

I would like to note especially that the authors added information on the vectors and theory on drivers of environmental predictors to the introduction, as well as added information on which variables were selected in the top performing models into the results. These additions satisfy my request for more content on this topic. I had also asked about an additional aim that would address these results, but I am happy with the rewriting of the aims and the explanation from the authors regarding their intent with this paper, given the variables have been further explained in other sections.

I would also like to thank the authors for the clarity regarding the region designations.

Thank you for clarification regarding variable selection using the NB model, and the additional sup figure illustrating how this choice did not impact the output.

I appreciated as well the sup fig S7 visualising the predictive performance on the validation datasets.

I understand the issue with data sharing and I appreciate the response. Supplying the code is an excellent step by the authors in any case and should be helpful for others hoping to do apply some of the models illustrated here to their own data.

I have one more detailed comment from this reading, and a few small notes below.

The overall conclusion from the paper is that the NB model fits the data better, and is "simple, stable and effective for prediction" (line 37). It strikes me that this statement comes from the combined inference from the Fig 4 model fit, and the predictive performance on the validation sets. The following statements seem to be drawn entirely from the relative heights of the bars in Fig 4 in the results section in lines 287-290: "Considering the linear and non-linear models have similar BIC values, there is no evidence that non-linear models fit the data substantially better. Over the three model components considered here, the over-dispersion effect is most important, followed by zero inflation, with non-linearity being the least important in fitting these data." The bars are are hard to distinguish visually. It is very hard to see the difference in value of the ZI Poisson models compared to the non ZI ones. Even the difference between the NB and P based models is not obvious. If this figure is carrying most of the weight of the main conclusion, I still think this needs to be more evident. I understand the author's comments about the practicality of a boxplot given these bars are averages of three values, but I find this visual unconvincing to carry such a large amount of the conclusions.

I would still recommend a few tweaks to Fig 1. Where it says “1. Split training sets and validation sets (Details in right)” > are you referencing the bottom right panel? If so, maybe put that panel above the Details of variable selection panel and say “… (Detail in TOP right)” and for step 2 say “… (Details in BOTTOM right)”. I would get rid of the colours in step three. Some of them replicate colours in the steps 1-5 of variable selection and that’s confusing.

Line 207 "assess"

Experimental design

no comment

Validity of the findings

no comment

Additional comments

no comment

---

## Round 0.3 · accepted · Accept

My apologies for the slow decision. I think you were aware that our thoughtful reviewer was on leave and it has also taken me longer than I would have liked to go through your latest manuscript.

As you can see below, the reviewer is happy with this version of your work and, having also reread it, I am delighted to accept your manuscript. I am looking forward to seeing it generate discussion amongst researchers in this, and I hope, other areas where similar choices sometimes arise.

I will make a few extremely minor suggestions for you to consider when you are finalising your manuscript for publication. These are about capitalisation, abbreviations, and hyphenation, rather than anything substantive, so please feel free to make use of them as you see appropriate.

Line 62: I’m not sure why the first word in “Hurdle models” is capitalised here. See also Line 199 and 202.

Line 62: Is there a possibility that adding the abbreviation “GAM” will help some readers recognise “generalised additive models”? Also, while I don’t think there is a rule here, I slightly more often see “Generalised Additive Models”, which would match the capitalisation you use elsewhere, e.g., Line 78’s “Negative Binomial”, but you are consistent with “generalised linear models”, also on Line 78.

Line 78: As above, you use “Negative Binomial” here (and on Lines 79 and 81), but usually don’t capitalise these words (e.g. Lines 37 and 192). I’d understand if this was when referring to the figures and supplementary tables using abbreviations, but that doesn’t seem the case here (c.f. Lines 208–213).

Line 200: It might be how I’m reading this, but this (“zero data can be structural zeros from the zero-part or sampling zeros from the non-zero-part of the model”) sounds like a single assumption (rather than plural assumptions) to me. Perhaps even “One of the assumptions of zero-inflated models is that…, which is a better conceptual fit to RRV data…”?

Line 201: As per above, I wonder if you’ve considered adding “conceptual” before “fit” here to make this point absolutely clear.

Line 207: I’m not sure about the capitalisation of the first word in “Standard Poisson” here (c.f Line 253). See also Line 308 (“standard negative binomial”) but then Lines 364–365 (“standard Negative Binomial”) and then Lines 408 and 419 (“standard negative binomial”).

Line 209: I’m not sure about the hyphen in “zero-part”. See also Lines 209 (again), 210, 211, etc.

·

Basic reporting

I would like to thank the authors for their work and patience. I appreciate the edits that have been made in response to the minor revisions. The methodology figure is very clear, and the breaks in the y axis I think are a big help to the reader.

Experimental design

NA

Validity of the findings

NA

Additional comments

NA